# An In Vitro Comparison of the Neurotrophic and Angiogenic Activity of Human and Canine Adipose-Derived Mesenchymal Stem Cells (MSCs): Translating MSC-Based Therapies for Spinal Cord Injury

**DOI:** 10.3390/biom10091301

**Published:** 2020-09-09

**Authors:** Ibtesam R. T. Al Delfi, Chelsea R. Wood, Louis D. V. Johnson, Martyn D. Snow, John F. Innes, Peter Myint, William E. B. Johnson

**Affiliations:** 1Dentistry Department, Kut University College, Al-hay Street, Kut, Waset, Iraq; ibtesamradhi@gmail.com; 2Faculty of Medicine, Dentistry and Life Sciences, University of Chester, Parkgate Road, Chester, Cheshire CH1 4BJ, UK; chelsea.wood@chester.ac.uk (C.R.W.); louisdvj@gmail.com (L.D.V.J.); 3Royal Orthopaedic Hospital, Birmingham B31 2AP, UK; martyn.snow@nhs.net; 4Veterinary Tissue Bank, Chirk L14 5ND, UK; info@vtbank.org (J.F.I.); peter.myint@vtbank.org (P.M.)

**Keywords:** mesenchymal stem/stromal cell, spinal cord injury, neurotrophic activity, angiogenic activity, translational research, regenerative medicine

## Abstract

The majority of research into the effects of mesenchymal stem cell (MSC) transplants on spinal cord injury (SCI) is performed in rodent models, which may help inform on mechanisms of action, but does not represent the scale and wound heterogeneity seen in human SCI. In contrast, SCI in dogs occurs naturally, is more akin to human SCI, and can be used to help address important aspects of the development of human MSC-based therapies. To enable translation to the clinic and comparison across species, we have examined the paracrine, regenerative capacity of human and canine adipose-derived MSCs in vitro. MSCs were initially phenotyped according to tissue culture plastic adherence, cluster of differentiation (CD) immunoprofiling and tri-lineage differentiation potential. Conditioned medium (CM) from MSC cultures was then assessed for its neurotrophic and angiogenic activity using established cell-based assays. MSC CM significantly increased neuronal cell proliferation, neurite outgrowth, and βIII tubulin immunopositivity. In addition, MSC CM significantly increased endothelial cell migration, cell proliferation and the formation of tubule-like structures in Matrigel assays. There were no marked or significant differences in the capacity of human or canine MSC CM to stimulate neuronal cell or endothelial cell activity. Hence, this study supports the use of MSC transplants for canine SCI; furthermore, it increases understanding of how this may subsequently provide useful information and translate to MSC transplants for human SCI.

## 1. Introduction

Much research has been performed to better understand the effects of spinal cord injury (SCI) and it sequelae, and how best to repair the damage caused and alleviate the loss of function. However, the absence of preclinical models that closely represent human SCI represents an important block in progress into potential SCI therapies. There is a clear need for the further development of SCI models that help bridge gaps between laboratory-based in vitro studies, preclinical small animal studies and human clinical trials [1,2,3].

Cell therapies for SCI require translational models to determine how they can be developed for clinical application. Rats have most commonly been used for such work [4], but, despite advantages in anatomical and mechanistic studies, these small animal models have a number of limitations, including physiological differences in neuroplasticity and in their capacity to recover compared with humans. For example, rats inflicted with a complete contusion SCI recover a capacity for weight-bearing and some movement within 1–2 weeks of injury, as determined by observational scoring, whilst humans can remain with a complete loss of motor function for life [5]. In addition, because SCI does not naturally occur in rodents, the injury must be induced in a controlled laboratory setting. This benefits analysis of specific neuronal defects, interventions and experimental reproducibility; however, it does not address the natural heterogeneity of human SCI, or the likely heterogeneity of new treatment outcomes [6,7].

Larger animals, such as cats, dogs and nonhuman primates, are at scale and physiologically more akin to humans than rodents [1,2,7]. Hence, many studies have examined the effects of cell transplantation after SCI in these larger models. Of these animals, the use of companion dogs in particular have been seen as advantageous as these animals naturally and often suffer from SCI as a result of degenerative disc disease and traumatic herniation of the intervertebral disc [8,9,10,11]. Furthermore, the histopathology and functional outcomes of canine SCI has been studied, with findings showing that SCI in dogs and humans is comparable [12,13]. Therefore, cell therapy research in naturally occurring SCI in dogs has been suggested to represent an important translational route to enable the prospects of developing successful new treatments for human SCI [8]. There is an emerging picture that comparison of the underlying biology in human SCI and naturally occurring canine SCI, and of potential therapeutics in both species, is worthwhile and may be mutually beneficial.

In recent years, transplantation of mesenchymal stem cells (MSCs) has developed as a promising candidate to abrogate the neural damage cause by SCI and help repair injured neural tissues [14,15,16]. The therapeutic effects of transplanted MSCs are likely due to their capacity to secrete neurotrophic, angiogenic, anti-inflammatory and immunomodulatory factors that prevent or overcome the effects of the nerve-inhibitory milieu in SCI lesion sites [15,17,18,19]. In a previous study, we reported that humans with complete SCI maintain a population of MSCs after long-term injury that can be cultured and increased in cell numbers [20]. Furthermore, we demonstrated that human [21,22] and canine [23] MSCs secrete factors into cell culture-conditioned medium (MSC CM) that have neurogenic and angiogenic effects. However, we have not directly compared this regenerative paracrine activity of MSCs derived from humans with those from dogs. Therefore, with a view to further informing the translation of MSC-based therapies for SCI from dogs to humans, and potentially vice versa, this study has compared the effects of human MSC CM and canine MSC CM on established models of neuronal and endothelial cell growth.

## 2. Materials and Methods

### 2.1. MSC Isolation and Growth

Ethical approval was gained for human MSC isolation from the excised infrapatellar fat pad of patients undergoing knee replacement surgery, Research Ethics Committee (REC) approval number 12/EE/0136; National Research Ethics Service (NRES) Committee, East of England, Hertfordshire. The tissue was obtained from donors with informed consent and the research was in accordance with the Declaration of Helsinki. Institutional approval was provided for canine MSC isolation from the University of Chester Faculty of Science and Engineering Research Ethics Committee: REC approval number 060/16/CW/BS. Canine cells were isolated from surgically extracted inguinal fat pads of dogs undergoing MSC transplantations for their treatment of joint pathology. Both the human MSC and canine MSCs were isolated by collagenase digestion as described previously [23], and cultured in T75 culture flasks with Dulbecco’s Modified Eagle Medium (DMEM)/F-12 culture medium supplemented with 10% fetal bovine serum (FBS) and 1% penicillin/streptomycin (standard culture medium; all from Gibco^®^, Life Technologies™, Paisley, UK). Cells were incubated at 37 °C in an atmosphere of 5% CO_2,_ 95% air.

### 2.2. Phenotyping Human and Canine MSCs 

MSC characterisation was undertaken according to the guidelines of the International Society for Cellular Therapy, including microscopic evidence of the cells being adherent to tissue culture plastic, demonstrating the capacity to differentiate along mesodermal lineages, and exhibiting a defined CD immunoprofile [24]. The plastic adherence of culture-expanded cells was determined by phase contrast microscopy, whilst the differentiation potential of MSC cultures was examined by treating cultures with established adipogenic, osteogenic and chondrogenic induction medium [20,23], as described below (minimum of n = 3 human and n = 3 canine donors):

#### 2.2.1. Adipogenic Differentiation

Cells seeded in standard culture medium at a density of 10^4^ cells/mL (1 mL/well) and allowed to adhere in 24-well culture plates were treated for four weeks with standard culture medium supplemented with 1 μM dexamethasone (Sigma Aldrich, Dorset, UK), 1% insulin, transferrin and selenium (ITS; Gibco^®^, Life Technologies™), 0.5 μM 3-isobutyl-1-methylxanthine (IBMX; Sigma) and 100 μM indomethacin (Sigma). Culture medium was replaced every 2–3 days. Control medium consisted of solvent carriers, i.e., methanol and dimethylsulfoxide (DMSO), at the same dilution used for the induction medium. After four weeks in culture, the cells were fixed and stained for the presence of lipid accumulation using Oil Red O staining [20,23].

#### 2.2.2. Osteogenic Differentiation

Cells seeded in standard culture medium at a density of 104 cells/mL (1 mL/well) and allowed to adhere in 24-well culture plates were treated for four weeks with standard culture medium supplemented with 10 μM beta-glycerophosphate, 50 μM ascorbic acid and 10 nM dexamethasone (all Sigma), with medium replaced every 2–3 days. Control medium contained solvent carriers, i.e., methanol and cell culture water at the same dilution used for the induction medium. After four weeks in culture, the cells were fixed and stained for the presence of the osteogenic marker, alkaline phosphatase activity [20,23].

#### 2.2.3. Chondrogenic Differentiation

Culture-expanded cells were harvested by trypsinisation and centrifugation and resuspended in DMEM/high glucose (Gibco^®^, Life Technologies™) supplemented with 2% FBS and 1% penicillin/streptomycin and aliquoted into 1.5 mL Eppendorf tubes (1ml/tube) at a final cell density of 2.5 × 10^5^ cells/mL. Following microcentrifugation at 500× *g* for 5 min, the cells formed small cell pellets, then the medium was replaced with DMEM/high glucose supplemented with 37.5 μg/mL ascorbate-2-phosphate (Sigma), 1% ITS, 100 nM dexamethasone and 10 ng/mL transforming growth factor beta-1 (TGF-β1; Peprotech, New Jersey, USA), with medium replaced every 2–3 days. Control medium contained the solvent carriers, i.e., ethanol, cell culture water and bovine serum albumin/phosphate-buffered saline (BSA/PBS, Sigma) at the same dilution used for the induction medium. After four weeks in culture, the pellets were fixed, embedded in paraffin and tissue sections stained for the presence of the chondrogenic marker, glycosaminoglycan deposition, using toluidine blue [20,23].

#### 2.2.4. CD Immunoprofiling

The CD profiles of human MSC cultures were examined at passage IV using immunolabelling and flow cytometry, as described previously [20]. Briefly, cells were harvested by trysinisation and centrifugation to form cell pellets and then resuspended at a density of 5 × 10^5^ cells/mL in 2% BSA/PBS. Initially, the cells were blocked with 10% normal human immunoglobulin (Ig) (Grifols, Cambridge, UK) for 1 hr. Cells were then washed with 1ml of 2% BSA/PBS and centrifuged (1000 rpm) to pellet before resuspending in 2% BSA/PBS at a minimum of 10^5^ cells/mL. One of the following phycoerythrin (PE) or fluorescein isothiocyanate (FITC) conjugated mouse monoclonal anti-human antibodies were added to aliquots: PE-CD34, FITC-CD44, PE-CD45, PE-CD105 (Immunotools, Friesoythe, Germany), PE-CD73 or PE-CD90 (BD Biosciences, Oxford, UK). The cells were also incubated with isotype-matched control antibodies IgG2a and IgG1 (Immunotools) to detect nonspecific fluorescence. The CD profiles of canine MSCs also were performed using immunolabeling and flow cytometry at passage IV, as described previously [23]. Aliquots of cells were resuspended in 200 μL 2% BSA/PBS, then incubated for 30 min at room temperature with 5 μL antibodies specific for canine antigens or validated for canine cross-reactivity against; PE-CD34 (BD Biosciences), FITC-CD44, PE-CD45, and PE-CD90 or IgG1 mouse, IgG2a rat IgG2b rat and IgG2b (all eBiosciences, ThermoFisher Scientific, Loughborough, UK) rat matched isotype control respectively. 

After immunolabelling, cells were washed with 2% BSA/PBS and immunoreactivity for each CD marker was performed by flow cytometry using a Beckman Coulter FC500 Flow Cytometer (Beckman Coulter (UK) Ltd., High Wycombe, UK) and data was analyzed using Kaluza^®^ Analysis Software.

### 2.3. Preparation of Conditioned Medium from Human and Canine MSC Cultures

The conditioned medium was prepared and harvested from human and canine MSCs, adapting previously established protocols for the determination of neurotrophic and angiogenic activity [21,22,23]. Briefly, human or canine MSC were seeded at 1.5 × 10^6^ cells in a T75 flask and left overnight for adherence. Flasks were then washed three times with sterile PBS and 15 mL of serum-free DMEM/F12 medium supplemented with 1% penicillin/streptomycin was added to each flask. After three further days of culture, conditioned medium (CM) was harvested, 0.2 μm filter sterilised (Minisart, Germany) and stored in aliquots at −80 °C until used. The viability of the MSCs cultures during this 72-h incubation remained greater than 95% (determined by trypan blue exclusion at day 3). Serum-free nonconditioned medium that had been incubated in T75 flasks, but with no cells seeded, was generated following the same procedure and used as a control in this study.

### 2.4. Assessment of the Neurotrophic Effects of Human and Canine MSC CM

SH-SY5Y neuroblastoma cells were used to examine the neurotrophic effects of human and canine MSC CM, as reported previously [21,23]. In brief, SH-SY5Y cells were seeded in 24-well plates in DMEM/F12 medium supplemented with 10% FBS and 1% penicillin/streptomycin at 2 × 10^4^ cells/mL/well and left to adhere overnight. Medium was removed and replaced with either human or canine MSC CM or control medium and placed in the Cell IQ live cell imaging platform (Cell-IQ version 2, CM Technologies Oy, Tampere, Finland) for three days at 37 °C/5% CO_2_. Cell IQ analyser software was used to measure total SH-SY5Y neuronal cell number and total neurite outgrowth in each treated condition. Further examination of neural differentiation was performed via βIII tubulin immunofluorescent staining. Briefly, SH-SY5Y cells seeded onto sterilised glass coverslips, allowed to adhere overnight, (i) gently washed in PBS and incubated in MSC CM or control medium for three days, (ii) fixed in 10% formalin (Sigma), then washed twice with PBS and blocked with 2% BSA/PBS for an hour, (iii) incubated at 4 °C for 2 h with the primary antibody, mouse monoclonal anti-βIII tubulin (Abcam, Cambridge, UK), (iv) washed with PBS and incubated in the dark at 4 °C for 1 h with secondary antibody, rhodamine Red-X-AffiniPure donkey anti-mouse IgG (H+L) (Stratech Ltd., Newmarket, UK), (v) after a final wash, the coverslips were mounted with Vectashield (Vectalabs, New South Wales, Australia) and imaged using a Leica DMI4000 digital microscope (Leica, London, UK).

### 2.5. Assessment of the Angiogenic Effects of Human and Canine MSC CM

EA.hy926 endothelial cells were used to determine the angiogenic effects of human and canine MSC CM, where scratch wound assays measured endothelial cell migration and proliferation (further determined by MTT assay) and the formation of endothelial tubule-like structures was examined using Matrigel assays, as reported previously [22,23]. 

For the scratch wound assays, EA.hy926 cells were seeded in standard culture medium into a 24-well plate and cultured in monolayers to confluence. Using a 200-μL pipette tip, each well was scratched centrally from top to bottom, leaving a gap void of cells, then the well was washed (to remove debris) prior to the additions of MSC CM or control medium. The plates were then incubated in Cell IQ live cell imaging platform for two days at 37 °C/5% CO_2_. The percentages of scratch closure wound area, endothelial cell migration, and the numbers of dividing cells were measured over this time course using the Cell IQ analyzer software. 

To assay endothelial tubule formation, a substrate of reduced growth factor Matrigel (Corning, USA) was established in 96-well plates then EA.hy926 cells were seeded at 2 × 10^4^ cells/well in 200 μL of either human or canine MSC CM or control medium and the plates incubated at 37 °C/5% CO_2_. After this time, total endothelial tubule length and the numbers of tubule branch points were measured using the Cell IQ platform and analyzer software.

### 2.6. Assessment of Viable Cell Proliferation Using MTT Assays

The proliferation of viable SH-SY5Y and EA.hy926 cells was examined following culture with human and canine MSC CM or control medium, for three and two days, respectively, using MTT assay (Sigma), as previously described [22,23].

### 2.7. Statistical Analysis

Data were generated from MSC cultures established from at least three human donors and three canine donors and have been reported as means ± standard error of the means (SEM). For statistical analysis, *p* ≤ 0.05 was considered as a significance threshold, which was determined using one-way or two-way ANOVAs with Tukey’s multiple comparison post-hoc test in GraphPad Prism 7 (GraphPad Software). 

## 3. Results

### 3.1. Phenotypic Characterisation of Human and Canine MSC Cultures 

The human and canine cells isolated and culture-expanded from adipose tissue fulfilled the criteria previously proposed by the International Society for Cellular Therapy (ISCT) to be considered MSCs [24]. These cell cultures consisted only of plastic-adherent cells with a fibroblastic appearance, which demonstrated the differentiation potential to form adipocytes, osteoblasts, and chondrocytes. After adipogenic induction, intracellular lipid droplet accumulation was confirmed by Oil Red O staining, whilst alkaline phosphatase activity was present when cells cultured in osteogenic induction medium, and the accumulation of glycosaminoglycans (GAGs) in the extracellular matrix of cell pellets were seen when cultured in chondrogenic induction medium, as indicated by toluidine blue staining (Figure 1a,b). Following analysis by flow cytometry, the human and canine cell cultures showed a lack of immunoreactivity for the hematopoietic CD markers, i.e., CD34 and CD45, while showing immunopositivity for the MSC-associated markers, i.e., CD44 and CD90, with each marker being detected in greater than 90% of all cell populations (Figure 1c).

### 3.2. The Neurotrophic Effects of Human and Canine MSC CM on SH-SY5Y Neuronal Cells

There was a marked increase in the numbers of SH-SY5Y neuronal cells present and in the extension of neurites from these cells when cultured for three days in human and canine MSC CM compared to those SH-SY5Y cells cultured in control medium (Figure 2a). Culturing SH-SY5Y cells in either human or canine MSC CM also increased βIII tubulin immunoreactivity compared to control conditions (Figure 2b). Automated image analysis was used to measure SH-SY5Y cell numbers, SH-SY5Y neurite outgrowth and the proportions of βIII tubulin immunopositive cells. This demonstrated that whilst both human and canine MSC CM significantly and markedly increased each of these parameters compared to the control medium, there were no significant differences between human versus canine MSC CM (Figure 2c). There were similarly increased absorbance levels detected by MTT assays of SH-SY5Y cell cultures in human and canine MSC CM compared to the control medium (Figure 2c, upper right panel). The MTT assay detects metabolic activity in viable cells. Given the increased numbers of SH-SY5Y cells seen in MSC CM this increased absorbance is supportive of the MSC CM increasing the numbers of viable cells present, although it may also indicate increased metabolic activity. 

### 3.3. The Angiogenic Effects of Human and Canine MSC CM on EA.hy926 Endothelial Cells

Assessment of the effects of MSC CM on EA.hy926 endothelial cells using automated image analysis of the scratch wound assay included three parameters; scratch wound closure, cell movement tracking and the numbers of dividing cells present over a 2-day period (Figure 3a). Human and canine MSC CM significantly promoted scratch wound closure when compared with the control medium (*p* ≤ 0.0001 and *p* ≤ 0.05, respectively). In addition, tracking individual EA.hy926 cell movement demonstrated that both human and canine MSC CM increased endothelial cell movements towards the centre of scratch area, when compared with the control medium (both *p* ≤ 0.0001). Furthermore, the number of dividing cells within the scratch area also showed significant increases when cultured in human and canine MSC CM compared with culture in the control medium (*p* ≤ 0.0001). MTT assays of EA.hy926 endothelial cell cultures indicated significantly increased absorbance levels in human and canine MSC CM compared to the control medium, which may be due to increased cell proliferation (as more dividing cells in scratch wound assay were seen by image analysis) and/or increased metabolic activity (Figure 3a, bottom right panel). There were no significant differences in any of these parameters between the effects of human MSC CM versus canine MSC CM. 

Assessment of EA.hy926 cells forming endothelial tubule-like structures in Matrigel assays was quantified using automated image analysis to count the total length of formed tubule-like structures and the total number of tubule branch points from digitised images. EA.hy926 cells cultured for 1 day in human and canine MSC CM had increased their formation of tubule-like structures compared to culture in control medium, where there were no tubule-like structures present (Figure 3b). There were significant increases in both the total tubule length and branch point formations in human and canine MSC CM when compared with control medium (both *p* ≤ 0.01 and *p* ≤ 0.05, respectively). There were no significant differences in these parameters in human MSC CM versus canine MSC CM.

## 4. Discussion

Currently, there is no cure for SCI, but recent research on cell-based therapies has shown encouraging results, promising improvement in some lost functional activity. MSCs are largely investigated both for their feasibility to treat SCI as an autologous cell transplant and because of their capacity to secrete an array of bioactive molecules, including growth factors and anti-inflammatory or immunomodulatory cytokines that prevent tissue damage and enhance wound healing [14,17,18,19]. There is, however, a clear need of an animal model that reflects specific problems of human SCI, including aspects of wound heterogeneity and scale-up issues, e.g., to determine therapeutic doses and optimised delivery of cell transplants. Other researchers have used natural SCI in dogs as a means to inform future cell therapies in humans, as well as developing new treatment options in veterinary medicine. Hence, SCI in dogs has been treated with autologous olfactory ensheathing cells [25] and MSCs [26], the latter study being completed by researchers who further trialed the treatment of human SCI with autologous MSC transplants [27] where both the canine and human recipients demonstrated improved functional outcomes. The emergence of naturally occurring SCI in companion dogs as a model for new treatments of humans with SCI is now recognised within the field [8,9,10,12].

In this study, we directly compared the neurogenic and angiogenic capacity of human adipose-derived MSCs with their canine counterparts to better understand similarities or differences between the two species. We found that adipose-derived human and canine MSCs have a similar phenotype that largely matches that proposed by the ISCT to be considered MSCs [24]. The extent to which canine MSCs underwent adipocyte differentiation was poor compared to human MSCs; however, this could be attributed to the induction medium, which followed a protocol established for human cells [20], and previously used in our laboratory for canine cells [23], rather than developed specifically for canine cells [28]. This method was to minimize laboratory-based experimental variation, but also because the focus of our research was to compare paracrine activity, not MSC differentiation potential. Importantly, we found that the neurogenic and angiogenic effects of MSC CM from human and canine MSC cultures were comparable and both increased SH-SY5Y neuronal proliferation, βIII tubulin immunoreactivity and neurite outgrowth, and EA.hy926 endothelial cell proliferation, migration and the formation of endothelial tubule-like structures, to a significantly greater extent than control medium, indicating marked trophic activity. The neurogenic [19,21] and angiogenic [22,29] effects of human MSC CM have been well documented, with much research showing that MSCs secrete neuroprotective, neurotrophic, angiogenic and immunomodulatory factors that are likely to inhibit the damage and increase repair after SCI [reviewed in 17]. Recently, a comprehensive proteomic determination of the content of canine MSC CM similarly demonstrated the secretion of a wide variety of trophic and immunomodulatory factors [30]. Furthermore, other research has shown that canine bone marrow MSCs secrete neurotrophic factors and angiogenic factors into the conditioned medium, which promoted angiogenesis using the chorioallantoic membrane assay [31]. Transplantation of canine MSCs into dogs with SCI also was associated with their expression of neurotrophic factors in vivo [32] and increased neural regeneration [33] and functional recovery after SCI [33,34,35]. In a recent study, injections with canine MSC CM was similarly associated with increased functional recovery in SCI dogs [36]. Hence, our data, combined with these earlier studies, support the hypothesis that canine MSCs are trophic cells that have application in wound healing and tissue regeneration, and hence a comparison of the paracrine activity of human and canine MSCs is useful when considering how these cells may function following transplantation after SCI. 

The current study suggests that the effects of human and canine MSCs are likely to be maintained across species. However, there is a caveat to this interpretation as we have used human responder cells only, i.e., SH-SY5Y neuronal cells and EA.hy926 endothelial cells, to examine the effects of human and canine MSC CM. This has some advantages in terms of the experimental consistency, but it is possible that the responses of these human cells to growth factors and other factors present in canine MSC CM do not accurately reflect the responses of canine neural and endothelial cells. Whilst it seems likely that, if anything, there would be enhanced neurotrophic and angiogenic effects seen if the canine MSC CM were applied to primary canine cells, it remains important to perform this analysis. Further research also is required to examine whether the comparative trophic effects of human and canine MSC CM seen in the cell-based assays reported herein accurately reflect what occurs in the more complex in vivo microenvironment of the damaged CNS, requiring animal models of SCI.

## 5. Conclusions

In conclusion, this study has examined the paracrine effects of human and canine MSCs cultured in identical conditions and using identical assays of neurotrophic and angiogenic activity. We have shown that MSCs from both species have a comparable phenotype and similar marked trophic effects on neuronal and endothelial cells. These findings may prove useful as further research examines the outcomes of future MSC-based therapies in dogs and humans with SCI.

## Figures and Tables

**Figure 1 biomolecules-10-01301-f001:**
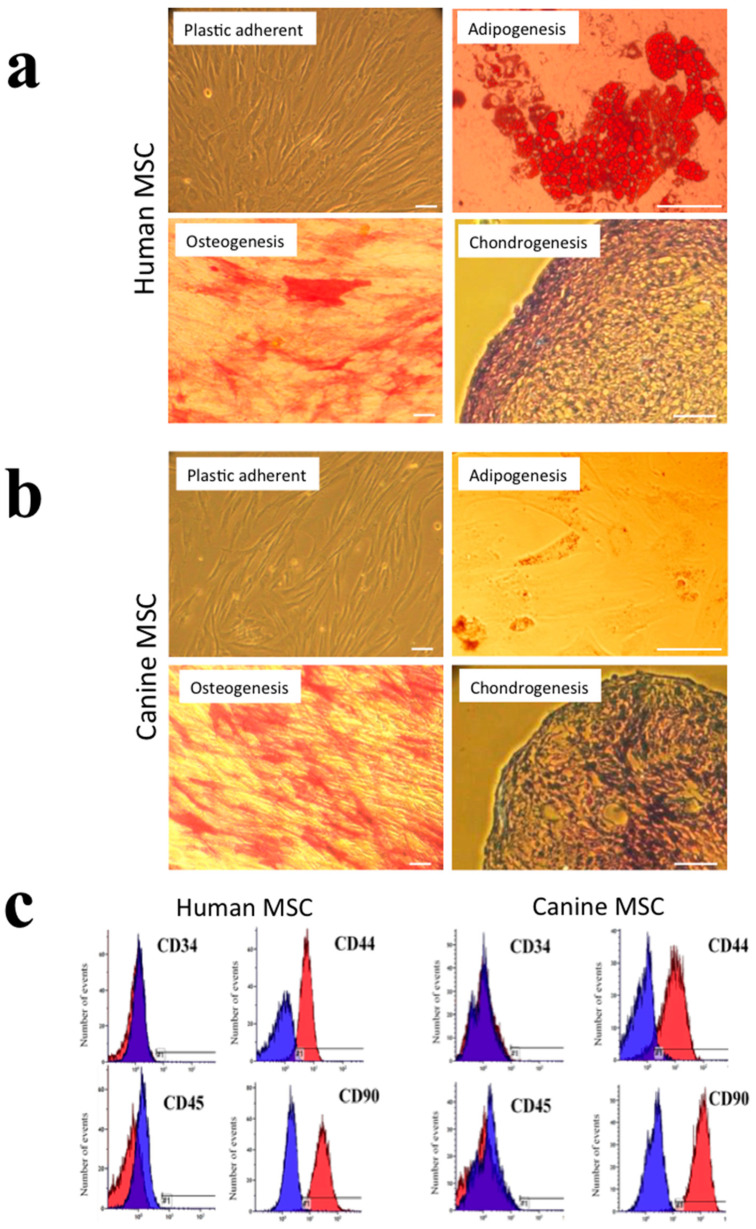
Phenotypic characterization of human and canine mesenchymal stem cells (MSCs). (**a**) Human MSCs were plastic adherent with a fibroblastic morphology under phase-contrast microscopy (top left panel). Following induction, human MSCs showed Oil Red O positivity, alkaline phosphatase positivity and secreted a toluidine blue-stained cartilaginous extracellular matrix, indicating their differentiation potential to form adipocytes, osteoblasts and chondrocytes, respectively (as indicated). (**b**) Canine MSCs exhibited a similar phenotype, although their propensity to differentiate to form Oil Red O positive adipocytes was less marked. Scale bars = 50 μm. (**c**) Representative histograms of flow cytometric analysis of human (top panels) and canine (bottom panels) MSCs following immunocytochemical staining for CD34, CD44, CD45 and CD90. Immunoreactivity with irrelevant isotype-matched control antibodies is shown in blue, while immunoreactivity for each of the CD markers is shown in red.

**Figure 2 biomolecules-10-01301-f002:**
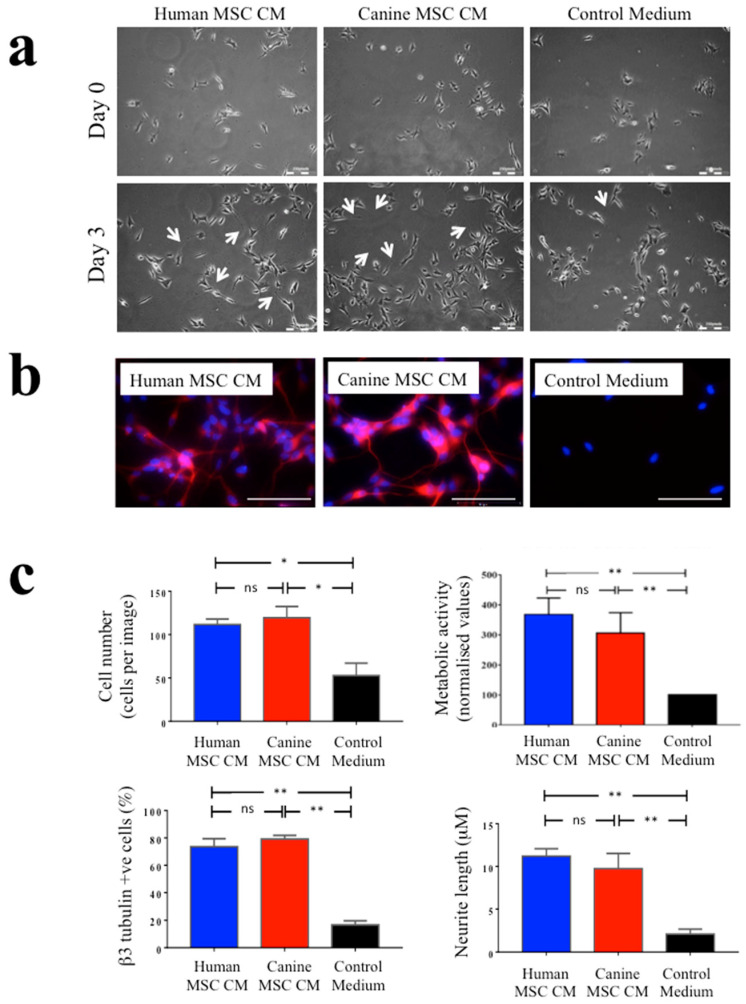
Human and canine MSC conditioned medium increased SH-SY5Y cell proliferation, βIII tubulin immunopositivity and neurite outgrowth to a similar extent. (**a**) Representative images are shown of SH-SY5Y neuronal cells under phase-contrast microscopy following culture for three days in human MSC conditioned medium (human MSC CM), canine MSC conditioned medium (canine MSC CM), or control medium. As shown, there was enhanced cell numbers and neurite outgrowth (arrowed) in both human and canine MSC CM compared with control cultures. Scale bars = 200 μm. (**b**) Representative fluorescence images of SH-SY5Y cells after three days of culture in human MSC CM, canine MSC CM or control medium following immunocytochemical staining for the neuronal marker, βIII tubulin. As shown, βIII tubulin immunopositive cells were seen in human and canine MSC CM to a much greater extent than in control conditions. (**c**) (top panels) Image analysis software demonstrated significantly increased SH-SY5Y cell numbers in the presence of human and canine MSC CM compared to control medium after three days in culture (top left) as well as increased metabolic activity as determined by MTT assays (top right); (bottom panels) there were significant increases the proportions of βIII tubulin immunopositive cells in human and canine MSC CM compared to the control medium, as well as increased neurite length. * *p* ≤ 0.05, ** *p* ≤ 0.01, ns = nonsignificant. Data shown as means ± SEM, with a minimum of three independent experiments.

**Figure 3 biomolecules-10-01301-f003:**
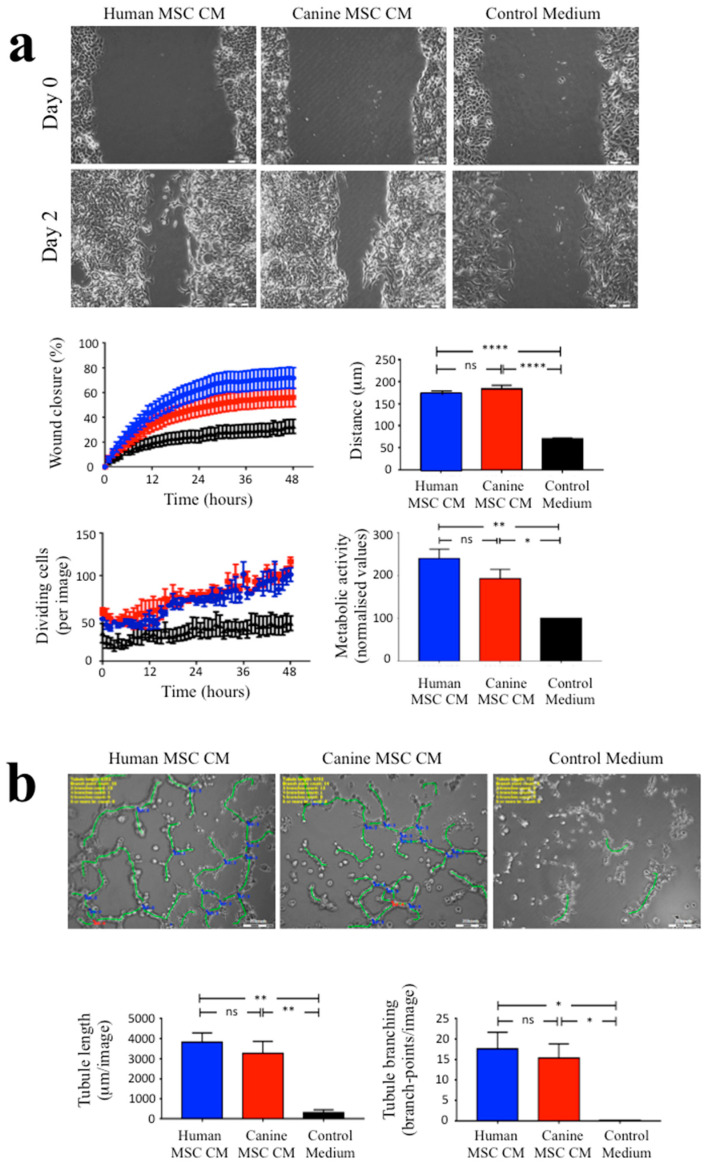
Human and canine MSC conditioned medium increased EA.hy926 endothelial cell migration and proliferation and induced the formation of endothelial tubule-like structures in Matrigel assays. (**a**) Representative images are shown under phase-contrast microscopy of EA.hy926 endothelial cell scratch wound assays. Image analysis of digitized images collected every 15 min over a two-day period demonstrated that scratch wound closure and endothelial cell migration was markedly greater in the presence of human and canine MSC CM versus control medium, with no significant difference seen between human and canine MSC CM (top graphs). Further image analysis (left lower panel) showed a significant increase in the numbers of dividing EA.hy926 cells in the presence of human and canine MSC CM compared to the control medium. In addition, there was increased metabolic activity in MSC CM treated cultures compared to control cultures (lower right panel). Scale bars = 200 μm. (**b**) Representative phase-contrast images are shown of EA.hy926 endothelial cells cultured on Matrigel in the presence of human MSC CM or canine MSC CM versus control medium. (Bottom panels) Significant increases were observed in both the total endothelial tubule length (per image) and the total number of endothelial branch points in EA.hy926 cultures treated with human and canine MSC CM compared to the control medium. For all panels, * *p* ≤ 0.05, ** *p* ≤ 0.01, **** *p* ≤ 0.0001, ns = nonsignificant. Scale bars = 200 μm. Data are shown as means ± SEM, with a minimum of three independent experiments.

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
