# Peer review of "An In Vitro Comparison of the Neurotrophic and Angiogenic Activity of Human and Canine Adipose-Derived Mesenchymal Stem Cells (MSCs): Translating MSC-Based Therapies for Spinal Cord Injury"

_biomolecules, 2020, doi:10.3390/biom10091301_

Round 1

Reviewer 1 Report

The authors present a comparison of conditioned media from human MSCs and canine MSCs in their effects on a neuronal cell line and an endothelial cell line. They show that CM treatment from either species leads to greater endothelial proliferation/migration in a scratch wound assay and greater proliferation/neuronal phenotypic signs in neuronal cells compared to control media. The CM from each species does not differ in efficacy in these assays. The studies are clear and straight-forward. They represent a sensible building block in the process of verifying canine MSCs as viable models of human MSCs. Only one minor correction is noted below:

Figure 1: The legend panel labeling is off. The canine MSCs lack a letter in the legend and the flow plots are mis-labeled as panel b (when they are panel c). Also, including labels on the figure itself indicating “human” and “canine” would improve readability.

Author Response

We are very grateful to the reviewer for their consideration of our work and for the helpful comments in addressing some errors in the Figure 1 legend. We have now corrected the figure legend to fit with the figure shown. We also have added the terms Human MSC and Canine MSC to the figure to improve its readability.

Reviewer 2 Report

Al Delfi et al. compared the effects of human and canine adipose-derived MSC CM on cell proliferation and wound-healing. Both MSC CM promote the proliferation of neuloblastoma and umbilical vein cell line. However, their conclusion is too exageratted. In addition, there are many studies that demonstrated neurotrophic and angigenic effects of MSCs.  Therefore, novelty is entirely low.

1. English editing is necessary. There are many mis-spelling and grammatical error.

2.  Why the authors could conclud MSCs have the neurtrophic activity from this experiment? I feel it is an exageration. MSC CM only increased neuloblastoma cell line.  Additional experiments are necesary.

3. It was reported in various studies that canine MSCs can promote the neuroregeneration. However, the authors never referred to them.

4. It is also an exageration that MSCs have angiogenic activity in resuls of Fig. 3.

Author Response

We are very grateful to the reviewer for their consideration of our work.

Response to Comment 1.

We have examined the manuscript for English usage and have corrected typographical errors, addressing a number of specific errors also pointed out by additional reviewers. Otherwise, we consider the manuscript well written.

Response to Comment 2.

We have concluded that the MSCs examined have neurotrophic activity based on the effects of the MSC CM in inducing significant increases in neurite extension combined with an increase in the immunopositivity for the neuronal marker β3 tubulin using a responding cell model system. We and many other researchers in the field have used the SH-SY5Y neuroblastoma cell line (published reports) in this manner to examine neurotrophic activity in this manner. We agree that further research using primary neural cells, or, better still, in vivo studies, would provide more concrete evidence of neurotrophic activity. To acknowledge this limitation in our current work, we have now added the following text to form the last paragraph of the Discussion section in the paper.

Lines 357-368. The current study suggests that the effects of human and canine MSCs is likely to be maintained across species. However, there is a caveat to this interpretation as we have used human responder cells only, i.e., SH-SY5Y neuronal cells and EA.hy926 endothelial cells to examine the effects of human and canine MSC CM. This has some advantages in terms of the experimental consistency, but it is possible that the responses of these human cells to growth factors and other factors present in canine MSC CM do not accurately reflect the responses of canine neural and endothelial cells. Whilst it seems likely that, if anything, there would be enhanced neurotrophic and angiogenic effects seen if the canine MSC CM were applied to primary canine cells, it remains important to perform this analysis. Further research also is required to examine whether the comparative trophic effects of human and canine MSC CM seen in the cell-based assays reported herein accurately reflect what occurs in the more complex in vivo microenvironment of the damaged CNS, requiring animal models of SCI.

Response to Comment 3.

We apologise for not including appropriate and sufficient reference to research that has examined the effects of canine MSC in SCI studies. We have now referred to previous research articles that have demonstrated increased neurotrophic expression in vivo and increased neural regeneration and functional recovery after canine MSC transplantation in canine SCI. This includes a very recent article (July 2020) in which canine MSC CM was injected into dogs with SCI. To properly acknowledge this prior research, we have included the following text in the Discussion section of the paper, adding the references needed.

Lines 349-353: Transplantation of canine MSCs into dogs with SCI also was associated with their expression of neurotrophic factors in vivo [32], and increased neural regeneration [33] and functional recovery after SCI [33-35]. In a recent study, injections with canine MSC CM similarly was associated with increased functional recovery in SCI dogs [36].

Response to Comment 4.

We have concluded that the MSCs have angiogenic activity based on the significant increases seen after MSC CM treatment in endothelial cell proliferation, cell migration and the formation of endothelial tubule-like structures in Matrigel assays. We and other researchers in the field have used these measures (published reports) to examine angiogenic activity. We agree that further research using primary endothelial cells and/or in vivo studies would provide more concrete evidence of angiogenic activity. To acknowledge this limitation in our current work, we have changed the last paragraph of the Discussion section in the paper, as detailed above (Response to comment 3). In addition, we have included detail of how other in vitro assays have been used to demonstrate the angiogenic effects of MSC CM, i.e., the chicken chorioallantoic membrane assay (lines 347-349).

Reviewer 3 Report

In general, it is a high-quality manuscript that provides significant results that compare CM MSCs obtained from human and canine adipose tissue. The results confirm that the MSC CM obtained from both tissues do not differ significantly and represent a valuable source for neuroplasticity and regeneration of endothelial structures. The manuscript is well written, the material and methods and results are sufficient and the discussion appropriate.

However, there are few parts that need to be corrected. Please, find my suggestions, comments and recommendations.

In the part: Phenotyping human and canine MSCs

Please include for each  differentiating protocol , type of plastic material you have used (for example 6 well plates, 24 well plates or tissue flasks) and number of cells  used for each differentiation strategy.

Especially, in the case of chondrogenic differentiation indicate how many cells was  used for small pelets- after microcentrifugation  -103- 128 lines

Briefly, cells were harvested by trysinisation- correct -131 line

Flasks were then washed three times with sterile PBS and serum-free medium was  added –line 155

Please, indicate the type of Medium used  (DMEM, MEM??)

In brief, SH-SY5Y cells were seeded in 24 well plates in standard culture medium -please, indicate the type of medium used  (DMEM, MEM??)  - 164 line

The MMT- correct-     assay detects metabolic activity in viable cells. Given the increased numbers of SH-SY5Y cells seen in MSC   243

After  minor revision I recommend to accept this manuscript for publication in BIOMOLECULS

Author Response

We are very grateful to the reviewer for their careful consideration of our work and positive response. We appreciate the need for some increased detail in the protocols and for spotting a spelling (abbreviation) error.

We have corrected the specific points raised as follows:

Specific reviewer comments (italics):

Reviewer comment: In the part: Phenotyping human and canine MSCs Please include for each differentiating protocol , type of plastic material you have used (for example 6 well plates, 24 well plates or tissue flasks) and number of cells used for each differentiation strategy. Especially, in the case of chondrogenic differentiation indicate how many cells was used for small pelets- after microcentrifugation -103- 128 lines

Response: These details have now been added as follows.

Lines 91-95: Both the human MSC and canine MSCs were isolated by collagenase digestion as described previously [23], and cultured in T75 culture flasks with Dulbecco’s Modified Eagle Medium (DMEM)/F-12 culture medium supplemented with 10% fetal bovine serum (FBS) and 1% penicillin/streptomycin (standard culture medium; all from Gibco®, Life Technologies™, Paisley, UK).

Lines 105-110: 2.2.1. Adipogenic differentiation: Cells seeded in standard culture medium at a density of 104 cells/ml (1ml/well) and allowed to adhere in 24-well culture plates were treated for four weeks with standard culture medium supplemented with 1µM dexamethasone (Sigma Aldrich, Dorset, UK), 1% insulin, transferrin and selenium (ITS; Gibco®, Life Technologies™), 0.5µM 3-isobutyl-1-methylxanthine (IBMX; Sigma) and 100µM indomethacin (Sigma). Culture medium was replaced every 2-3 days.

Lines 113-117: 2.2.2. Osteogenic differentiation: Cells seeded in standard culture medium at a density of 104 cells/ml (1ml/well) and allowed to adhere in 24-well culture plates were treated for four weeks with standard culture medium supplemented with 10µM beta-glycerophosphate, 50µM ascorbic acid and 10nM dexamethasone (all Sigma), with medium replaced every 2-3 days.

Lines 121-124: Culture-expanded cells were harvested by trypsinisation and centrifugation and re-suspended in DMEM/high glucose (Gibco®, Life Technologies™) supplemented with 2% FBS and 1% penicillin/streptomycin and aliquoted into 1.5 ml Eppendorf tubes (1ml/tube) at a final cell density of 2.5x105 cells/ml.

Reviewer comment: Briefly, cells were harvested by trysinisation- correct -131 line (132)

Response: Further details have now been added as follows.

Lines 135-137. Briefly, cells were harvested by trysinisation and centrifugation to form cell pellets and then re-suspended at a density of 5x105 cells/ml in 2% BSA/PBS.

Reviewer comment: Flasks were then washed three times with sterile PBS and serum-free medium was added –line 155. Please, indicate the type of Medium used (DMEM, MEM??)

Response: Further details have now been added as follows.

Lines 159-160. Flasks were then washed three times with sterile PBS and 15ml of serum-free DMEM/F12 medium supplemented with 1% penicillin/streptomycin was added to each flask.

Reviewer comment: In brief, SH-SY5Y cells were seeded in 24 well plates in standard culture medium -please, indicate the type of medium used (DMEM, MEM??) - 164 line

Response: Further details have now been added as follows. Lines 168-170: In brief, SH-SY5Y cells were seeded in 24 well plates in DMEM/F12 medium supplemented with 10% FBS and 1% penicillin/streptomycin at 2x104 cells/ml/well and left to adhere overnight. Reviewer comment: The MMT- correct- assay detects metabolic activity in viable cells. Given the increased numbers of SH-SY5Y cells seen in MSC 243 Response: This has now been corrected to MTT (on line 248).